# Dietary Supplementation with Probiotic *Bacillus licheniformis S6* Improves Intestinal Integrity via Modulating Intestinal Barrier Function and Microbial Diversity in Weaned Piglets

**DOI:** 10.3390/biology12020238

**Published:** 2023-02-02

**Authors:** Wenjuan Sun, Wenning Chen, Kun Meng, Long Cai, Guiguan Li, Xilong Li, Xianren Jiang

**Affiliations:** 1Key Laboratory of Feed Biotechnology of the Ministry of Agriculture and Rural Affairs, Institute of Feed Research, Chinese Academy of Agricultural Sciences, Beijing 100081, China; 2COFCO Feed Co., Ltd., Beijing 100020, China

**Keywords:** *Bacillus licheniformis*, intestinal epithelial barrier function, intestinal integrity, intestinal microbiota, weaned piglets

## Abstract

**Simple Summary:**

The livestock industry urgently needs alternatives to antibiotics, especially in post-weaning piglets. Numerous studies support the notion that dietary supplementation probiotics seem to be one of the most promising tactics to reduce post-weaning diarrhea in pigs. *Bacillus li-caniforms* S6 (*BL*−S6) supplementation in piglets was investigated for its effect on growth performance and gut health. The researchers found that *BL*−S6 supplementation modulated the piglet’s immunity-oxidative capacity and composition of the cecum microbiota, which would, in turn, modulate intestinal barrier function, and eventually improve growth performance and relieves diarrhea. We highlight the potential role of *BL*−S6 as an option to improve growth performance and relieve diarrhea in pig production.

**Abstract:**

*Bacillus licheniformis* (*B. Licheniformis*) has been considered to be an effective probiotic to maintain gut health and boost productivity in the pig industry, but there is no complete understanding of its mechanisms. We determined whether weaned piglets exposed to *BL*−S6 (probiotic) had altered intestinal barrier function or microbiota composition. In our study, 108 weaned piglets (54 barrows and 54 gilts) were divided equally into three groups, each with six pens and six piglets/pen, and fed a basal diet supplemented without or with antibiotic (40 g/t of Virginiamycin and 500 g/t of Chlortetracycline) or probiotic (1000 g/t of *B. Licheniformis*) for a 14-day trial. On day 14, one piglet was chosen from each pen to collect blood and intestinal samples. Compared with the control group, dietary supplementation with a probiotic promoted body weight (BW) gain and average daily gains (ADG) while reducing diarrhea incidence (*p* < 0.05). Probiotics enhanced superoxidase dismutase (SOD) activity and decreased malondialdehyde (MDA) levels in serum (*p* < 0.05), and increased the level of mRNA expression of *SOD1*, *Nrf2,* and *HO-1* (*p* < 0.05) in the jejunum mucosa. Moreover, supplementation with probiotics improved intestinal mucosal integrity as evidenced by higher villus heights and a higher ratio of villus heights to crypt depths (duodenum and jejunum) and higher mRNA and protein levels of occludin and ZO-1 in jejunum mucosa (*p* < 0.05). The intestinal sIgA levels (*p* < 0.05) were elevated in the probiotic group, and that of serum immunoglobulin A (IgA) tended to be higher (*p* = 0.09). Furthermore, weaning piglets who were given probiotics had a better balance of the cecum microbiota, with *lactobacillus* abundance increased and *clostridium_sensu_stricto_1* abundance decreased. In conclusion, dietary supplementation with the probiotic *BL*−S6 promoted intestinal integrity, which was associated, in part, with modulating intestinal barrier function and microbial diversity in weaned piglets; it may offer a promising alternative to antibiotics to prevent diarrhea.

## 1. Introduction

In the swine industry, post-weaning diarrhea is a very serious issue due to the mortality, morbidity, and weight loss associated with it [1]. Antibiotics have been widely used in animal husbandry, providing benefits in improving body weight and reducing the incidence of post-weaning diarrhea in the past few decades. However, the misuse of antibiotics in diet can have serious consequences, including the initiation of antimicrobial resistance, loss of drug effectiveness, and adverse effects on people’s health [2,3]. Consequently, some countries completely ban antibiotics from livestock feed for production purposes, especially in the EU and China. Consequently, any other reliable strategy that could be used as an alternative to antibiotics will be needed to keep pigs healthy, especially in the post-weaning period. Antibiotic alternatives such as probiotics, plant extract, and acidifiers have gained researchers’ attention due to their ability to minimize antimicrobial drug resistance.

With the growing interest in probiotics and their use in animal studies, numerous studies support the concept that dietary supplementation with probiotics appears to be an effective way to relieve post-weaning diarrhea in pigs [4,5,6,7]. There is growing evidence indicating that probiotic supplementation can maintain gut health in pigs by altering the gut microbiota, enhancing immune regulation and barrier function, increasing nutrient digestibility, and consequently improving growth performance [4]. In comparison with other types of probiotics, *B. licheniformis* possesses obvious advantages for its spore-forming characteristics, which make it thermostable when confronted with long-term storage and feed processing and strong acid resistance at low pH in the stomach [8,9]. These above characteristics make *B. licheniformis* an ideal candidate for feed additives. Supplementation with *B. licheniformis* has been shown to reduce diarrhea incidence and severity, thus improving growth performance in piglets [10]. Specifically, *B.-licheniformis*-improved growth performance was ascribed to stimulating appetite, promoting digestion, increasing digestibility, and nutrient retention [11,12]. In a weaned piglet experiment, 1 × 10^9^ CFU/kg *B. licheniformis* was added to the basal diet; the results indicated that *B. licheniformis* helps to improve antioxidant capacity, promotes immune function, and regulates the intestinal microflora of weaned piglets [10]. In addition, *B. licheniformis* supplementation promoted the ratio of fecal *lactobacillus* in growing-finishing pigs as well as reduced their toxic gas emissions [11].

However, despite some biological functions of *B. licheniformis* having been identified, the detailed mechanism and the effect of different subtypes of *B. licheniformis* on intestinal health needs further study.

We hypothesized that dietary supplementation of *BL*−S6 modulates the intestinal microbial community and immune-oxidative status; ultimately, this would improve growth performance and gut health in weaned piglets by modulating the microbiome–gut axis. Therefore, we aimed to elucidate the effects of *BL*−S6, a potential alternative to the antibiotic, on growth performance and gut health, and furthermore, decipher the relationship among immune-oxidative status, microbiota composition, intestinal barrier function, and their potential effects on the weaned piglets’ gut health and growth performance.

## 2. Materials and Methods

### 2.1. Experimental Animals and Dietary Treatments

A total of 108 Duroc × Landrace × Yorkshire crossbred piglets (54 barrows and 54 gilts) were weaned at 21 ± 1 d of age (average initial BW 6.50 ± 0.21 kg), divided into 3 treatments (6 replicate pens/treatment, 6 piglets/pen according to BW and sex), and provided *ad* libitum access to feed and water. The 3 treatments included: a control group, an antibiotic group (a basal diet containing 40 g/t of virginiamycin and 500 g/t of chlortetracycline), and a probiotic group (basal diet + 1 × 10^11^ CFU/kg *BL*−*S6*). Feed was given for 14 days. The test strain *BL*−S6 used in the present study was screened and preserved in the laboratory. The dry powder product of *BL*−S6 contained live bacteria at a concentration of 1 × 10^11^ CFU/g. Diets for piglets were formulated based on the National Research Council (2012) and detailed in Appendix A.

### 2.2. Growth Performance and Diarrhea

The procedures of this research were adapted from a previous study [13]. Throughout the study, we score each pig for diarrhea by visual inspection daily score on a scale of 1 to 5 (1 = hard, dry pellet; 2 = firm, formed stool; 3 = soft, moisturizing, and holding shape; 4 = soft, unformed stool; and 5 = pourable water liquid). Diarrhea frequency was calculated as the percentage of days with a diarrhea score of 4 or higher. The BW and feed intake in each group were weighed on days 0 and 14. The ADG and average daily feed intake (ADFI) were calculated. Feed conversion rate (FCR) is the specific value of feed intake to weight gain.

### 2.3. Sample Collections

On day 14, one piglet, with a BW close to the average weight of the pen, was selected from each pen and then slaughtered and sampled. Serum was obtained by blood sampling from the jugular vein before slaughter. Intestinal samples were collected from piglets near the middle of the intestine (including duodenum, jejunum, and ileum); segments approximately 2~3 cm in length were fixed in 4% paraformaldehyde for analysis of intestinal morphology. The jejunum (approximately 10 cm in length) was dissected longitudinally for mucosa collection, which was snap-frozen in liquid nitrogen and then stored at −80 °C prior to further analysis. Chyme from the cecum was collected and snap-frozen in liquid nitrogen for extraction of fecal microbial genomic DNA. 

### 2.4. Digestive Enzyme Activity

Intestinal mucosal digestive enzyme activity was assayed by use of commercial kits (Amylase, REF, C010-1-1; Chymotrypsin activity, REF, A080-3-1; Lipase activity, REF, A054-1-1; Trypsin, REF: A080-2-1, Shanghai Enzyme-linked Biotechnology Co., Ltd., Shanghai, China).

### 2.5. Morphological Analysis of Small Intestine

Haematoxylin and eosin (H&E) staining were used to analyze intestinal morphology following the protocol used in our laboratory [13]. Digital photos of intestinal morphology were obtained through a microscope (Olympus CX31, Olympus Corporation, Tokyo, Japan), and 6 fields were randomly selected to measure the villus height and the crypt depth.

### 2.6. Enzyme-Linked Immunosorbent Assay

Commercial kits (Shanghai Enzyme-linked Biotechnology Co., Ltd., Shanghai, China) were used to analyze secretory immunoglobulin A (sIgA, Cat # ml603477) in jejunum mucosa and the content of immunoglobulin in the serum, including immunoglobulin A (IgA, Cat # ml660941), immunoglobulin G (IgG, Cat # ml002328), and immunoglobulin M (IgM, Cat # ml002334). The specific test procedures follow the manufacturer’s protocol.

### 2.7. Relative Quantitative Real-Time PCR(qRT-PCR)

Approximately 0.1 g of frozen jejunal mucosa samples were ground into powder and lysed in Trizol. Then, total RNA was reverse-transcribed with iScript cDNA Synthesis Kit (Bio-Rad, Hercules, CA, USA). qRT-PCR was carried out to determine the mRNA expression levels using a PCR Detection System (Bio-Rad, Hercules, CA, USA) and SYBR Green reagents (Bio-Rad, Hercules, CA, USA) under the manufacturer’s instruction. The specific primers were listed in Appendix A. GAPDH was used as the reference gene. By using 2^−ΔΔCt^, the expression differential of target genes is calculated relative to their reference genes [14].

### 2.8. Western Blot Analysis of Intestinal Mucosa Tight Junction Proteins

The protein of jejunum mucosa used for western blotting (WB) was prepared using RIPA lysis buffer added with protease inhibitor cocktail, and the protein concentrations were detected with the BCA Protein Assay Kit. In brief, aliquots containing 35 mg protein were separated by 12% SDS-PAGE and then transferred to PVDF membranes. Incubation with the primary antibody was performed overnight after the membrane was blocked in 5% skim milk for 1 h at room temperature. For immunoreactive bands, membranes were incubated with a second antibody conjugated with peroxidase (HRP), and chemiluminescence substrate (Thermo Fisher Scientific Inc., San Diego, CA, USA) was used after 3 washes. Bio-Rad Quantity One software (Bio-Rad Laboratories, Richmond, CA, USA) was used for the analysis of blot images. Appendix A lists dilution details for all antibodies.

### 2.9. Analysis of Microbiota in Cecum Digesta

Microbial diversity of cecal digesta was detected according to Majorbio’s standard protocol. The V3–V4 hypervariable regions of 16S rDNA were amplified with universal primers 338F (ACTCCTACGGGAGGCAGCAG) and 806R (GGACTACHVGGGTWTCTAAT). Illumina MiSeq PE300 platform (San Diego, CA, USA) was used to sequence PCR products. The sequences of raw data were quality-filtered with fast (0.19.6). Following the de-noising of the high-quality sequences, amplicon sequence variants (ASVs) were obtained. The number of reads from each sample was thinned out to 4000, which is still good with an average yield of 97.90%. ASV-based data analysis was performed on the Majorbio cloud platform (Bio-Pharm, Technology Co., Shanghai, China) according to the standard protocol, an online platform provided by Majorbio Biopharm Technology Co., Ltd. (Shanghai, China).

### 2.10. Measurement of Organic Acid in Cecum Digesta

Cecum digesta were analyzed for concentrations of microbial metabolites following previously reported protocol^3^. Sample analysis was performed by ion-exclusion chromatography on an ion chromatograph (IC; Metrohm, Switzerland) using a Metrosep Organic Acids-250/7.8 column. The mobile phase was 0.5 mM sulfuric acid and the column flow rate was 0.8 mL/min. Detection of organic acids was carried out using suppressed conductivity detection.

### 2.11. Statistical Analysis

SPSS 17 (SPSS Inc., Chicago, IL, USA) was used to analyze the data using a completely randomized block design. For growth performance, the pen was the experimental unit, and for intestinal parameters and serum parameters, each piglet was the experimental unit. Growth performance was measured by the pen, and intestinal parameters and serum parameters were measured by the piglets. Using Tukey’s test, multiple comparisons of treatment were made, and the incidence of diarrhea was analyzed using the chi-square test. Values are indicated as means ± SEM, and statistical significance was determined at *p* < 0.05 and tendency at 0.05 ≤ *p* < 0.10, respectively.

## 3. Results

### 3.1. BL-S6 Supplementation of Weaning Piglets’ Diets Affects Growth Performance and Diarrhea

Table 1 shows that ADG, ADFI, and FCR of the antibiotic group were not different from those of the control group, and FCR was not different among the three treatment groups. However, the supplementation of the probiotic increased BW on d 14, ADG, and ADFI compared to the control and antibiotic groups (*p* < 0.05).

Antibiotics and probiotic supplementation decreased diarrhea incidence in weaned pigs from days 0 to 14 compared with the control group (*p* < 0.05; Table 1).

### 3.2. Effect of Dietary BL-S6 Supplementation on Digestive Enzyme Activity

When dietary supplements with antibiotics and probiotics were given to piglets, chymotrypsin activity increased in the jejunum mucosa (*p* < 0.05; Figure 1B). Nevertheless, the activity of trypsin, lipase, and amylase activities among the three treatments were not significantly different (Figure 1A,C,D).

### 3.3. Dietary BL-S6 Supplementation Improves the Immune Antioxidant Status of Serum and Jejunum Mucosa

Although serum MDA levels were significantly lower in the probiotic group than in the control group (*p* < 0.05; Figure 2A), SOD activity was significantly higher in the probiotic group (*p* < 0.05; Figure 2B), but no differences were detected in the activity of catalase and glutathione peroxidase (GSH-px) (*p* > 0.05; Figure 2C,D). On day 14, antibiotics significantly increased serum IgA, IgG, and IgM concentrations compared with control (*p* < 0.05; Figure 2E–G), whereas probiotics showed a trend towards increasing IgA (*p* = 0.09; Figure 1E).

The effect of the probiotic on the gene expression levels of the antioxidant-related genes in the jejunum mucosa are shown in Figure 2H–N. Adding antibiotics/probiotics to piglets’ diets could increase SOD1, Nrf2, and HO-1 mRNA expression levels compared to controls (*p* < 0.05); however, a significant difference in CAT1, Gpx4, Keap1, and NOQ-1 did not exist among the three treatments (*p* > 0.05). In addition, immune effector factors, such as secretory immunoglobulin A (sIgA) and mucin-2 were also detected in the jejunum mucosa of piglets (Figure 1O,P). It was found that antibiotics/probiotics significantly increased sIgA levels in comparison to the control group (Figure 2O), but no differences were detected in mucin-2 mRNA expression levels (Figure 2P).

### 3.4. A Diet Containing BL-S6 Improves Weaned Pigs’ Intestinal Morphology and Epithelial Barrier Function

Figure 3A presents the results regarding intestinal morphology. In the duodenum, dietary supplement of probiotics increased the villus height and villus height/crypt depth ratio (*p* < 0.05), while antibiotics and probiotics did not significantly differ (*p* > 0.05). As compared with the control, dietary supplementation with the probiotic significantly increased the villus height/crypt depth ratio in the jejunum (*p* < 0.05).

According to Figure 4, tight junction proteins are present in weaned piglets’ jejunal mucosa. In the probiotic group, ZO-1 mRNA expression levels were significantly higher than those in the antibiotic group (*p* < 0.05; Figure 4B); probiotics increased occludin levels significantly compared to controls (*p* < 0.05; Figure 4C). Comparatively to controls, probiotics and antibiotics increased occludin and ZO-1 expression levels (*p* < 0.05; Figure 4D–F).

### 3.5. Microbiota Diversity in the Cecum Digesta in Response to BL-S6

Weaned pigs exposed to *BL*−S6 had altered cecum microbiota as shown in Figure 5. Within the three groups, 2139 core ASVs were detected, of which 522, 367, and 422 unique CSV were unique to the control, antibiotics, and probiotic groups, respectively (Figure 5A). For the analysis of β-diversity, using the first two principal component factors (31.41% and 22.28) of PC1 and PC2, PCoA plots based on Bray–Curtis distances were generated to demonstrate the differences in the composition of the three groups of microbes (Figure 5B). Alpha diversity was assessed according to Shannon, Simpson, Ace, and Chao indices; alpha diversity parameters did not differ among the three groups (Figure 5C–F).

### 3.6. The Effects of BL-S6 on the Bacterial Abundance in the Cecum Digesta

A diagram showing the composition and abundance of the cecum digesta microbiota is shown in Figure 6. Firmicutes and Bacteroidota dominate at the phylum level. The majority of dominant genus-level groups are *prevotella, lactobacillus*, and *clostridium_sensu_stricto_1* (Figure 6B). By evaluating LDA (LDA threshold > 4.0), the effect of microbial abundance on different effects was examined. Probiotic supplementation had an impact on the species-level composition of the cecum microbiota (Figure 6C,D). An increased richness of *s_unclassified_g_clostridium_sensu_stricto_1* in the control group, *s_uncultured_bacterium_g_anaerostipes* in the antibiotics group, as well as *s_unclassified_g_lactobacillus* and *s_uncultured_bacterium_g_alloprevotella* in the probiotic group were detected (Figure 6D). Probiotics showed an increased relative abundance of g_*lactobacillus* (*p* < 0.05) in the Kruskal–Wallis sum-rank test (Figure 6E); in contrast, the *g_clostridium_sensu_stricto_1* value decreased (*p* = 0.06) in comparison with the controls (Figure 6F). We conclude that dietary supplementation with the probiotic improves cecum microbiota balance by boosting *lactobacillus* relative abundance while reducing *clostridium_sensu_stricto_1* relative abundance.

### 3.7. Effects of BL-S6 on the Contents of Organic Acid in Cecum Digesta

Table 2 exhibited the effects of *BL*−*S6* supplementation in the diet on the composition of organic acid of the cecum digesta in weaning piglets. Probiotics increased lactate (*p* = 0.05), formic acid (*p* = 0.053), and propionic acid (*p* = 0.07) contents compared to the control and antibiotic groups, respectively, while the tendency of isobutyric acid was higher in the probiotic group when compared with the antibiotic group (*p* = 0.059). Acetic acid, butyric acid, isovaleric acid, and valeric acid content was not different among the three groups.

## 4. Discussion

The use of probiotics in animal husbandry has attracted widespread attention because of its potential to replace the use of antibiotics in feed to improve gut health and growth performance. Probiotics, such as *lactobacillus* and *bacillus*, have been shown to benefit animal growth [10,15,16]. Weaned piglets receiving *B. licheniformis* had improved growth, reduced post-weaning diarrhea, and improved intestinal epithelium and gut microbiota. [5,7,10,17]. Dietary supplementation of 10^9^ CFU/kg or 1.5 × 10^9^ CFU/kg *B. licheniformis* significantly improved growth performance in pigs and chickens, respectively [10,15]. The outcome of this study showed that dietary supplementation with antibiotics or *BL*−S6 (1 × 10^11^ CFU/kg) reduced the diarrhea incidence of weaned piglets, while it increased the growth performance of piglets; the effect of *BL*-S6 with a lower level on weaning piglets will be tested in our future study.

Redox status affects livestock growth and gut health, according to researchers [18,19,20]. Early weaning triggers redox imbalance, and total antioxidant capacity (T-AOC), SOD, GSH-Px, and MDA are important biomarkers reflecting the imbalance of redox status [21,22,23,24]. The T-AOC balances active oxygen, while MDA indicates how much oxidative damage has been done by ROS and lipid peroxidation, SOD serves to catalyze the conversion of reactive superoxide anions into hydrogen peroxide, and GSH-Px inactivates peroxide. *B. licheniformis* and bioactive compounds have been reported to enhance the oxidation capacity of pigs [10,25,26]. In piglets, the supplementation of *B. licheniformis* enhanced antioxidant capacities by increasing T-AOC, GSH-Px, and SOD levels, while reducing MDA levels [10]. The antioxidant capability was also significantly increased in goldfish-fed diets with *B. licheniformis*, increasing the antioxidant-related gene expression (CAT and GSR) [27]. This study also showed that *B. licheniformis* supplementation increased SOD activity and reduced MDA levels in serum. Furthermore, *BL*−S6 supplements are also found to increase the expression of antioxidant-related genes (Nrf2, SOD1, and HO-1) in weaned piglets’ jejunum mucosa.

Intestinal epithelial barriers can be damaged by weaning stress, harming gut health and growth performance [28]. Besides acting as a protective barrier, the intestinal epithelium facilitates nutrient absorption, and its morphology can be used to assess the function and upgrowth of the gut. Pigs’ intestinal barrier function was improved after the supplementation with *B. licheniformis* [5,29]. An increased villi height of the duodenum and the villi height/crypt depth of the duodenum and jejunum were reported after supplementation with *BL*−S6. (Figure 2). Accordingly, mucosal digestion enzyme activities and villi height were found to be linearly related in weaned piglets [30]. Bacillus increased the activity of the digestive enzyme in rabbits [31], and the activity of Chymotrypsin was elevated with the supplementation of dietary *BL*−S6 in this study, which might contribute to the improvement of intestinal function. Furthermore, tight junction proteins, including ZO-1, occludin, and claudin-1, play an important role in the maintenance of intestinal integrity and barrier function. Livestock exposed to probiotics (including *B. licheniformis*) express higher levels of tight junction proteins in the intestinal epithelium [6,32]. A dietary supplement containing *BL*−S6 significantly enhanced both the mRNA and protein content of intestinal epithelial tight junction proteins (ZO-1 and occludin) (Figure 3). The mucus layer is an important chemical barrier for intestinal epithelial cells against bacterial infection. Previous studies showed that *B. licheniformis* significantly increased intestinal mucin 2 in chickens and pigs [33,34,35]. However, our data demonstrated that *BL*−*S6* supplementation doesn’t affect the level of *mucin-2* which is consistent with the previous results that probiotics did not affect the intestinal mucin expression [36]. Intestinal mucin proteins are regulated by probiotics in a multi-faceted manner. The results from this study enable us to gain a better understanding of *BL*−S6’s beneficial effects on the intestinal epithelial barrier function of weaned piglets, which could lead to improved gut function.

Probiotics act as immune modulators to enhance the mucosal barrier of the host, thus preventing the intestinal epithelium from being infected by pathogens [37,38]. Immunoglobulin (IgA, IgM, IgG) is mainly existed in serum, which is one of the main components of the immune system in animals. Several studies describe the immunomodulatory effects of *B. licheniformis* [10,15,39]. Supplementation with *B. licheniformis* has been reported to improve immune function in piglet serum by increasing IgA, IgM, and IL-10 levels and reducing IL-6 and IL-1β levels [10]. Dietary supplementation of *BL*−*S6* could boost the non-specific immunity of *eriocheir sinensis* in *Chinese mitten crab eriocheir sinensis*; this may be regulated by the elevated expression of genes encoding immune-related enzymes in E. Sinensis blood cells [39,40]. Our study suggests that dietary supplementation with *BL*−*S6* may improve intestinal epithelial immune barrier function by increasing serum IgA concentration and sIgA content in jejunal mucosa. The above data indicated that *BL*-S6 supplementation contributes to intestinal mucosal immunity mediated by sIgA and to anti-infection immunity mediated by IgA. Recent studies reported that some probiotics (including *lactobacillus* [37,41,42,43], *bifidobacterium bifidum* [44], and *saccharomyces cerevisiae* [45]) promote enhancing the host sIgA abundance and other probiotics (*bacillus subtilis* [10,46]. In addition, probiotic bacteria, including *lactobacillus* [47,48], regulate IgA levels in the pig’s serum or IgA^+^ cells. Therefore, our findings contribute to a better understanding of the immune system’s regulation by probiotics.

Previous studies evidenced that probiotics contributed to promoting the colonization of the gut by beneficial bacteria, which was vital for improving the host’s health and growth performance [5,10,49,50]. Several studies have demonstrated the gut microbiota-modulatory effects owing to *B. licheniformis* supplementation [5,10,22]. Recent studies have shown that supplementing a mixture including *B. licheniformis* in pig diets significantly increases the Simpson diversity index of the gut microbiota [5]; however, our data demonstrated that *BL*−*S6* supplementation does not affect the parameters of α-diversity according to Shannon, Simpson, Ace, and Chao. Consistent with previous research results, Firmicutes and Bacteroidota were the predominant bacterial phyla in this study [10]. A genus-level alteration of gut microbiota was detected in the *BL*−S6 supplementation group, *lactobacillus* abundance was increased, and *clostridium_sensu_stricto_1* abundance was decreased following *BL*−*S6* supplementation. Numerous studies support the concept that the benefit of Lactobacillus to its hosts is far greater than the harm, which mechanisms mainly involved regulating antioxidant capacity, improving immune function and microbial homeostasis, reducing harmful bacterial colonization, etc. [51,52]. An analysis by Wang et al. (2021) found a negative correlation between *clostridium_sensu_stricto_1* level and ADG in piglets and litter weight gains [53]. However, some researchers revealed that the *B. licheniformis* group had an elevated abundance of *clostridium_sensu_stricto_1* compared with the control, which could reduce the diarrhea ratio by producing volatile fatty acids (VFAs), promoting the adherence of pathogens to the intestinal mucus barrier, decreasing the contents of inflammatory factors [10,48,54,55]. Numerous studies reported *B. licheniformis* may have distinct influences on the manipulation of intestinal microbiota, even the probiotics belonging to the same genus/species. Several factors influence the composition of gut microbial communities, including the level and subtype of probiotics, the feed formula, and the feeding mode. Therefore, the effects and mechanism of *B. licheniformis* on microbiota required further research.

## 5. Conclusions

A diet supplemented with probiotic *BL*−S6, a new subtype of *B. licheniformis*, improved piglets’ gut health and growth performance in weaned piglets. Furthermore, we demonstrated that *BL*−*S6* promoted intestinal integrity, which was associated, in part, with modulating intestinal barrier function and microbial diversity, thus providing a new option for us in pig production to prevent diarrhea.

## Figures and Tables

**Figure 1 biology-12-00238-f001:**
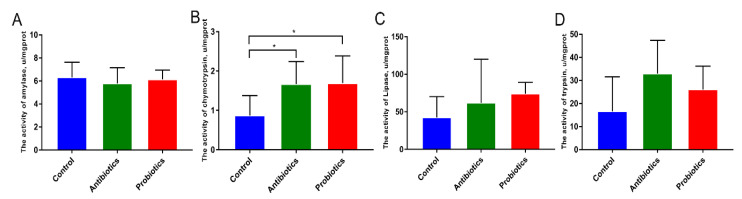
*BL*−S6 effects on digestive enzyme activity in piglets’ jejunum: (**A**) Amylase activity; (**B**) Chymotrypsin activity; (**C**) Lipase activity; (**D**) Trypsin activity. *n* = 6. * indicates the degree of significant difference (*p* < 0.05).

**Figure 2 biology-12-00238-f002:**
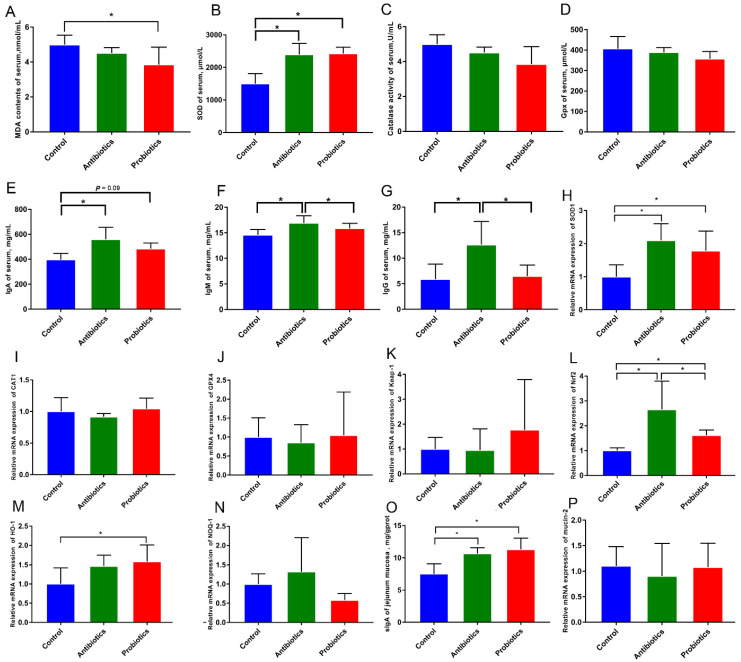
The effects of *BL*−S6 on antioxidant and immune status in serum and jejunum mucosa of piglets. (**A**) Serum Malondialdehyde (MDA) level. (**B**–**D**) Measuring SOD, CAT, and GSH-Px in serum. (**E**–**G**) The evaluation of serum IgA, IgM, and IgG in serum. (**H**–**N**) The mRNA expression level of SOD1, CAT1, Gpx4, Keep-1, Nrf2, HO-1, and NOQ. (**O**) The contents of sIgA in jejunum mucosa. (**P**) The mRNA expression level of mucin-2 in jejunum mucosa. *n* = 6. * indicates the degree of significant difference (*p* < 0.05).

**Figure 3 biology-12-00238-f003:**
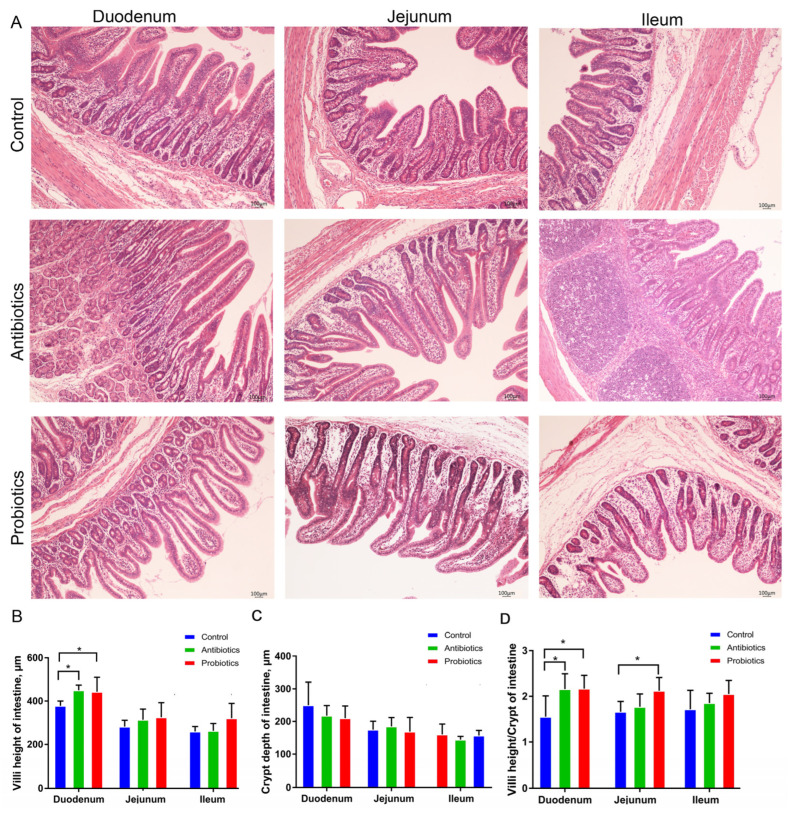
*BL*−S6 affects intestinal (duodenum, jejunum, and ileum) morphology of piglets. (**A**) H&E staining shows intestinal morphology (100×). (**B**,**D**) Statistical analysis of villus height (mm) (**B**), crypt depth (mm) (**C**), and the ratios of villus height (mm) to crypt depth (mm) (**D**) in the intestinal tract, respectively. * indicates the degree of significant difference (* *p* < 0.05). *n* = 6.

**Figure 4 biology-12-00238-f004:**
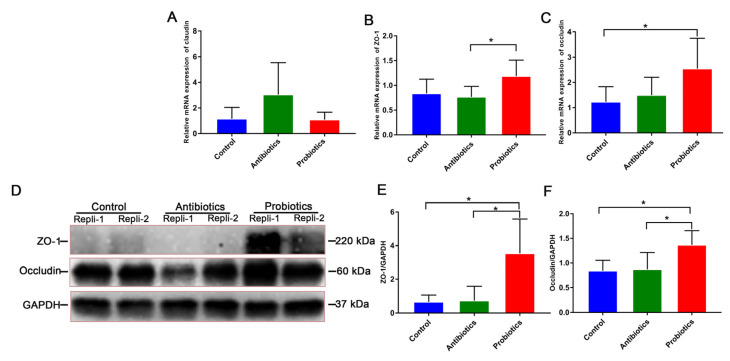
*BL*−S6 affects pigs’ physical barrier functions of intestinal epithelial. (**A**–**C**) The mRNA expression level of tight-junction protein in jejunum mucosa: claudin (**A**), ZO-1 (**B**), and occludin (**C**). (**D**–**F**) WB detects the protein levels of ZO-1, occludin, and a housekeeping protein (GAPDH) in the jejunum mucosa from weaned piglets. (**D**) Specific protein bands detected by WB; each lane represents an individual replicates; results of replicates 3~6 of each treatment are exhibited in Appendix A of WB original bands, replicate-1,2 (Repli-1,2). Semi-quantitative analyses of ZO-1 (**E**) and occludin (**F**). *n* = 6. * indicates the degree of significant difference (*p* < 0.05).

**Figure 5 biology-12-00238-f005:**
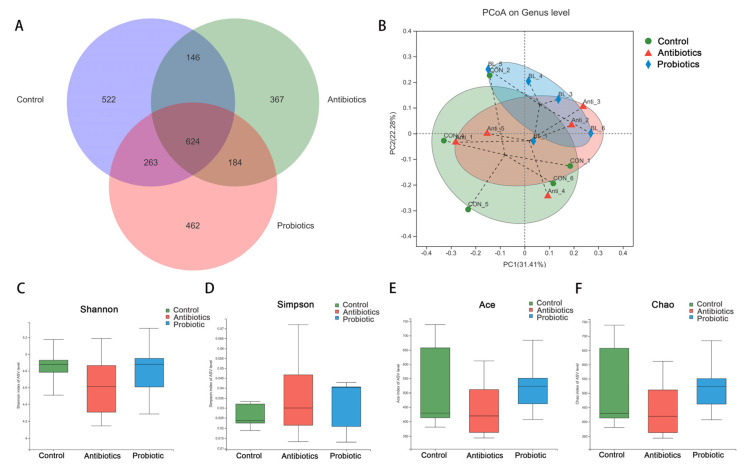
Effects of *BL*−S6 on microbiota diversity in cecal digesta of piglets. (**A**) Venn diagram of ASV in the cecum digesta. (**B**) Plots of PCoA based on Bray–Curtis distances. (**C**–**F**) The Shannon, Simpson, Ace, and Chao index. Values are presented as mean ± SEM, *n* = 5.

**Figure 6 biology-12-00238-f006:**
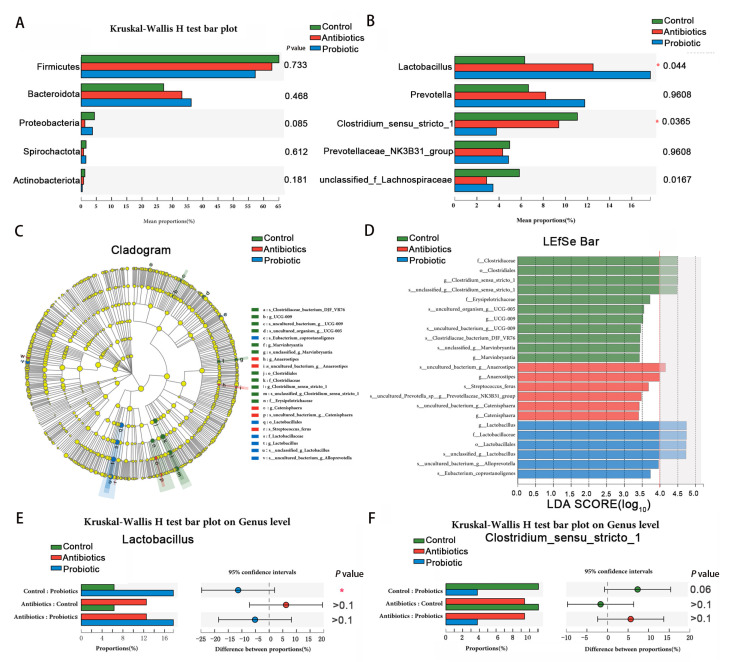
Piglet cecum microbiota is affected by *BL*−*S6*. (**A**,**B**) Test for significance at the phylum and genus level among treatments. (**C**,**D**) Discriminant analysis of multi-level species differences by LEfSe analysis from phylum to species level. Changes in *lactobacillus* (**E**) and *clostridium_sensu_stricto_1* (**F**) at the genus level. *p* values < 0.05 were considered significant, and *p* values between 0.05 and 0.10 were considered a tendency, *n* = 5. * indicates the degree of significant difference (*p* < 0.05).

**Table 1 biology-12-00238-t001:** Growth performance and diarrhea incidence of weaned pigs fed *BL*−S6 enriched diets.

Items	Control	Antibiotics	Probiotic	SEM	*p* Value
Weight, kg					
Day 0	6.50	6.50	6.50	0.21	0.97
Day 14	8.1 ^b^	7.94 ^b^	8.52 ^a^	0.24	<0.001
ADG, g	115 ^b^	103 ^b^	144 ^a^	5.2	<0.001
ADFI, g	195 ^ab^	182 ^b^	231 ^a^	7.7	0.02
FCR	1.71	1.77	1.60	0.03	0.21
Diarrhea incidence, %					
Day 0–14	14.85 ^a^	7.34 ^b^	10.71 ^b^	-	0.001

ADFI = average daily feed intake; ADG = average daily gain. ^a,b^ Mean with different superscripts in the same row differ significantly (*p* < 0.05).

**Table 2 biology-12-00238-t002:** The contents of organic acid (mg/kg) in cecum digesta of weaned pigs with diets supplemented with *BL*−*S6*.

Items	Control	Antibiotics	Probiotic	SEM	*p* Value
Lactate	150 ^b^	153 ^b^	468 ^a^	162	0.024
Formic acid	15.8 ^y^	21.2 ^xy^	28.6 ^x^	6.7	0.064
Acetic acid	5552	5256	4972	312	0.215
Propionic acid	1315 ^y^	1273 ^xy^	1541 ^x^	185	0.062
Isobutyric acid	105 ^xy^	91 ^y^	118 ^x^	12	0.068
Butyrate	1076	1008	914	214	0.921
Isovaleric acid	93	66	78	16	0.393
Valeric acid	199	188	220	26	0.200

^a,b^ Means listed in the same row with different superscripts are significantly different (*p* < 0.05). ^x,y^ Means listed in the same row with different superscripts tended to be different (0.05 ≤ *p* < 0.10).

## Data Availability

There are datasets available from the current study.

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
