# Peer review of "Dietary Supplementation with Probiotic Bacillus licheniformis S6 Improves Intestinal Integrity via Modulating Intestinal Barrier Function and Microbial Diversity in Weaned Piglets"

_biology, 2023, doi:10.3390/biology12020238_

Round 1
Reviewer 1 Report
In this manuscript, the authors investigated the effect of supplementing pigs' diet with Bacillus licheniformis (BL-S6). They look for changes in growth performance, the incidence of diarrhea, digestive enzymes activity, antioxidant and immune status (antibodies), intestinal morphology, barrier function, and microbiota. The authors conclude that a diet supplemented with probiotic BL-S6 improved piglets' intestinal integrity. I found it interesting, with a good number of animals that can provide statistical significance. However, I have some minor points that I would like to be addressed.
1) please correct the spelling of the scientific names for Bacillus licheniformis. The correct nomenclature requires lowercase for the species name, I found it written with capital letter in several parts of the text. The same for other scientific names
2) Authors measure the digestive enzyme activity in Figure 1; however, other than describing the results (no changes), they do not discuss why their a need for that study and how this help to address the research question.
3) Figure 4D show western blot for zonulin-1 and occluding, they show 3 treatments but there are 6 lanes, the figure showed 2 lanes for each treatment, why is that? are they replicates? different animals?. I believe each lane should be described
4)Figures 5 and 6.- the letters are way too small, and difficult to read, even if zoomed in. could lettering be increased to facilitate reading?
5) Add a brief concluding remark as authors contribution as new knowledge in the field since B. licheniformis has been extensively used in the pig's industry, and there are several reports on growth performance, microbiota changes, immune function, and even antioxidants actitvity
6) I provided a file that shows typos and other concerns through the text

Author Response
Review 1
In this manuscript, the authors investigated the effect of supplementing pigs' diet with Bacillus licheniformis (BL-S6). They look for changes in growth performance, the incidence of diarrhea, digestive enzymes activity, antioxidant and immune status (antibodies), intestinal morphology, barrier function, and microbiota. The authors conclude that a diet supplemented with probiotic BL-S6 improved piglets' intestinal integrity. I found it interesting, with a good number of animals that can provide statistical significance. However, I have some minor points that I would like to be addressed.
Response (R): Thank you very much for the careful evaluation and thoughtful comments to improve the quality of our manuscript, and we revised the manuscript and marked up using the “Track Changes” in the text.
1) Please correct the spelling of the scientific names for Bacillus licheniformis. The correct nomenclature requires lowercase for the species name, I found it written with capital letter in several parts of the text. The same for other scientific names
R: Thank you very much for the reviewer’s comment. We modified the spelling of the species name for “Bacillus licheniformis” into “bacillus licheniformis” in the revised manuscript, and the same correction was made for other scientific names in the revised manuscript.
2) Authors measure the digestive enzyme activity in Figure 1; however, other than describing the results (no changes), they do not discuss why their a need for that study and how this help to address the research question.
R: Thank you very much for the suggestion, and we added the corresponding discussion in the revised manuscript. Please check Lines 335-339.
3) Figure 4D show western blot for zonulin-1 and occluding, they show 3 treatments but there are 6 lanes, the figure showed 2 lanes for each treatment, why is that? are they replicates? different animals? I believe each lane should be described
R: Thank you very much for the reminder to avoid confusion in Figure 4D. We described each lane in Figure 4D, each lane represents a sample from an individual animal. Since the gel used in the present study has 10 lanes, so a total of 18 samples from 3 treatments could not be contained in one gel. Thus, 2 lanes for each treatment in Figure 4D represent replicate 1 and replicate 2, and the sample of replicate 3~4 and 5~6 was contained separately in another two gels. Please check the results exhibited in the supplementary files.
4) Figures 5 and 6.- the letters are way too small, and difficult to read, even if zoomed in. could lettering be increased to facilitate reading?
R: We increased the size of the letters in Figures 5 and 6 according to your suggestion.
5) Add a brief concluding remark as authors contribution as new knowledge in the field since B. licheniformis has been extensively used in the pig's industry, and there are several reports on growth performance, microbiota changes, immune function, and even antioxidants actitvity.
R: We agree that there are some previous studies concerning the efficacy of B. licheniformis in pigs. However, there are some differences between our current study and previous studies. Firstly, BL-S6 is a new subtype of B. licheniformis which can improve piglets' gut health and growth performance in weaned piglets, thus providing a new option for us in pig production to prevent diarrhea. Secondly, we demonstrated that BL-S6 promoted intestinal integrity, which was associated, in part, with modulating intestinal barrier function and microbial diversity in weaned piglets. Please check Lines 400-404.
6) I provided a file that shows typos and other concerns through the text.
Reply: Thank you very much for your kind corrections, and we modified the relevant typos and concerns in the revised manuscript.
Reviewer 2 Report
Although the approach is not very original, this paper is clearly presented, English is correct, the results are discussed clearly and supported by a well-documented bibliography.
I have few comments to make.
Line 34 the serum immunoglobulins A (IGA) levels "tended" to be higher
Line 52 I think it should be good to mention also the EU that was the first to ban the AGP....
Line 96 it should be good to mention clearly the final concentration in feed i.e. 1 x 1011 cfu/kg.... Could you clarify also if this is for spores or for spores + vegetative cells ?
Line 195 : Unless I am mistaken, the method of measuring digestive enzymes is not mentioned in materials and methods
Line 307 Compared to the studies you mention it seems that the concentrations used here are relatively high 1x1011 CFU/kg. Can you comment on this? Would bacillus licheniformis E6 levels at 109 cfu/kg be as effective?
Author Response
Review 2
Although the approach is not very original, this paper is clearly presented, English is correct, the results are discussed clearly and supported by a well-documented bibliography.
I have few comments to make.
Response (R): Thank you very much for your comments. We modified the manuscript according to your corrections and comments.
Line 34 the serum immunoglobulins A (IGA) levels "tended" to be higher
R: Thank you very much for the suggestion. We added the word. Please check Line 36.
Line 52 I think it should be good to mention also the EU that was the first to ban the AGP.
R: We modified “especially in the USA and China” into “especially in the EU and China” according to your suggestion. Please check Line 53.
Line 96 it should be good to mention clearly the final concentration in feed i.e. 1 x 1011 cfu/kg. Could you clarify also if this is for spores or for spores + vegetative cells?
R: Thank you very much for the suggestion. We modified “(basal diet + 1000g/t BL-S6)” into “(basal diet + 1 x 1011 CFU/kg BL-S6)” in the revised manuscript, and the concentration of BL-S6 is for spores. Please check Line 95.
Line 195: Unless I am mistaken, the method of measuring digestive enzymes is not mentioned in materials and methods
R: Thank you very much for the reminding. We added the method of measuring digestive enzymes. Please check Lines 120-123.
Line 307 Compared to the studies you mention it seems that the concentrations used here are relatively high 1x1011 CFU/kg. Can you comment on this? Would bacillus licheniformis E6 levels at 109 cfu/kg be as effective?
R: Thank you very much for the suggestion. In our study, we concluded that bacillus licheniformis S6, a new subtype of bacillus lichen forms, was effective on the gut health of weaning piglets as the supplemented level at 1×1011 CFU/kg. A previous study reported a certain subtype of bacillus licheniformis E6 at the level of 109 CFU/kg was effective, thus the effect of bacillus licheniformis S6 with a lower level at 109 CFU/kg on weaning piglets will be tested in our future study. Please check Lines 310-313.